# MINOCA: One Size Fits All? Probably Not—A Review of Etiology, Investigation, and Treatment

**DOI:** 10.3390/jcm11195497

**Published:** 2022-09-20

**Authors:** Lucas Lentini Herling de Oliveira, Vinícius Machado Correia, Pedro Felipe Gomes Nicz, Paulo Rogério Soares, Thiago Luis Scudeler

**Affiliations:** Instituto do Coração (InCor), Hospital das Clínicas, Faculdade de Medicina, Universidade de São Paulo, São Paulo 05403-900, Brazil

**Keywords:** MINOCA, myocardial infarction, coronary artery disease

## Abstract

Myocardial infarction with non-obstructive coronary arteries (MINOCA) is a heterogeneous group of conditions that include both atherosclerotic (coronary plaque disruption) and non-atherosclerotic (spontaneous coronary artery dissection, coronary artery spasm, coronary artery embolism, coronary microvascular dysfunction, and supply–demand mismatch) causes resulting in myocardial damage that is not due to obstructive coronary artery disease. Failure to identify the underlying cause may result in inadequate and inappropriate therapy in these patients. The cornerstone of managing MINOCA patients is to identify the underlying mechanism to achieve the target treatment. Intravascular imaging is able to identify different morphologic features of coronary plaques, while cardiac magnetic resonance is the gold standard for detection of myocardial infarction in the setting of MINOCA. In this review, we summarize the relevant clinical issues, contemporary diagnosis, and treatment options of MINOCA.

## 1. Introduction

Patient is admitted to the emergency department with chest pain, ischemic electrocardiogram (ECG), and highly positive troponin. The patient is sent to the cath lab, but no coronary obstructions were detected.

Myocardial infarction with nonobstructive coronary arteries (MINOCA) is defined as an acute myocardial infarction (AMI) without significant coronary artery obstruction on angiography (>50%) or without specific imaging findings.

MINOCA was first described by Gross and Steinberg in 1939 [1]. Recent studies have found a prevalence of MINOCA of 2–11% in AMI patients [2,3,4,5,6]. However, prevalence varied widely across the studies.

The prognosis of patients with MINOCA is far from benign. Patients with MINOCA are at increased risk for adverse cardiovascular events such as AMI and death [3,7,8]. However, it is difficult to establish an accurate prognostic assessment of patients with MINOCA, as the prognosis can be influenced by the cause, as well as the degree of myocardial damage associated with AMI. 

MINOCA may present with or without ST-segment elevation and, in general, patients have lower increases in cardiac troponin than patients with obstructive coronary artery disease [9,10].

A myriad of conditions can lead to MINOCA, and the mechanisms involved are both atherosclerotic and non-atherosclerotic, although the underlying cause of AMI is not always apparent. Therefore, cardiovascular imaging tests have a critical role in assessing patients with MINOCA.

This review aims to explore the relevant clinical issues, contemporary diagnosis and treatment options of MINOCA.

## 2. Diagnostic Criteria

The term MINOCA refers exclusively to ischemic conditions, such as epicardial vasospasm and non-obstructive atherosclerotic plaque instability. However, several non-ischemic diseases have similar presentations to MINOCA. Therefore, the term MINOCA is frequently used as a descriptive diagnosis until further evaluation confirms an ischemic mechanism or unravels an alternate diagnosis, such as Takotsubo cardiomyopathy (TCM) or myocarditis. 

Recently, the European Society of Cardiology [11] and the American Heart Association [12] defined MINOCA according to specific criteria (Table 1).

Therefore, the diagnostic criteria are very useful in exposing the ischemic nature of MINOCA. 

## 3. Epidemiology

Recent studies have found a prevalence of MINOCA of 2% to 11% in AMI patients [2,3,4,5,6].

Both acute myocardial infarction with obstructive coronary artery disease (AMI-CAD) and MINOCA are more prevalent in men. However, women are more likely to present with MINOCA than men, representing 24% of the AMI-CAD population and 43% of the MINOCA population [9]. 

Patients with MINOCA tend to be somewhat younger than AMI-CAD patients, with a median age at presentation of 58 years [9]. The prevalence of traditional cardiovascular risk factors seems similar in both groups, with the exception of hypercholesterolemia, which is less frequent in MINOCA patients [9]. 

The Variation in Recovery: Role of Gender on Outcomes of Young AMI Patients (VIRGO) study evaluated the clinical characteristics and outcomes of young patients (age between 18 and 55 years) with MINOCA versus AMI-CAD [2]. The registry showed an incidence of MINOCA of 11%. MINOCA was more common in women and less associated with traditional risk factors than AMI-CAD (8.7% versus 1.3%; *p* < 0.001). Moreover, similarly to the data presented by Pasupathy et al. [9], the VIRGO study showed that MINOCA patients presented less frequently with ST-segment-elevation myocardial infarction (STEMI) compared to patients with AMI-CAD (21% vs. 52%, *p* < 0.001). 

The prognosis of patients with MINOCA is not benign, but the long-term mortality after MINOCA is lower than that in patients with AMI-CAD. Annual mortality varies from 1.15% to 3.5% [7,9]. Other studies have shown an increased risk of new AMI and hospitalization for heart failure [13,14]. 

Pelliccia et al. showed that reduced left ventricular ejection fraction, nonobstructive coronary artery disease, use of beta blockers during follow-up, and ST-segment depression on the admission electrocardiogram are significant predictors of long-term prognosis in patients with MINOCA [7]. 

## 4. Physiopathology

MINOCA is a heterogeneous entity with various mechanisms responsible for the acute presentation. The adequate mechanism by which ischemia happens is a temporary suspension of blood flow to the myocardium, which usually takes place in the epicardial arteries, but which may also happen in the microvasculature [15]. 

The American Heart Association 2019 statement categorized the causes of MINOCA as atherosclerotic or non-atherosclerotic [12]. Atherosclerotic causes encompass plaque disruption while non-atherosclerotic causes encompass epicardial coronary vasospasm, coronary microvascular dysfunction, spontaneous coronary artery dissection, and supply-demand mismatch.

## 5. Atherosclerotic Causes

### Coronary Plaque Disruption

Studies have shown that approximately 38–40% of patients with MINOCA have some evidence of plaque disruption, including plaque rupture, erosion, or calcified nodules when intracoronary imaging is performed [16,17]. 

Plaque rupture is defined as fibrous cap discontinuity leading to a communication between plaque cavity and the coronary lumen [18]. Plaque erosion is defined as a thrombus contiguous to the luminal surface of a plaque without signs of rupture [19]. Calcified nodule is defined based on optical coherence tomography (OCT) imaging criteria as a signal-poor region with poorly delineated borders that protrudes into the arterial lumen [20].

An autopsy series of 800 cases of sudden coronary death found a prevalence of plaque rupture of 55–60%, plaque erosion of 30–35%, and plaque erosion of 2–7% [21]. 

This mechanism of coronary plate disruption can lead to in situ obstruction or to embolization of atherosclerotic debris and platelet aggregates, causing AMI. The coronary angiography does not evidence obstructive lesions possibly because of the endogenous fibrinolytic system [22] or due to the superimposed vasospasm on an unstable plaque, with normalization by the time of the catheterization (Figure 1). However, the angiographic appearance may suggest plaque disruption; for example, haziness or a small filling defect [12]. However, intravascular imaging can accurately diagnose plaque disruption, preferably with OCT or, to a lesser extent, with intravascular ultrasound (IVUS).

## 6. Non-Atherosclerotic Causes

### 6.1. Spontaneous Coronary Artery Dissection (SCAD)

The first case of SCA was described by Pretty in 1931 [23]. Since then, SCAD continues to be misdiagnosed, underdiagnosed, and incorrectly managed, which may harm patients with SCAD.

SCAD is defined as a separation of the layers of an epicardial coronary-artery wall by intramural hemorrhage, with or without an intimal tear [24]. This intramural hemorrhage may progress in a way that it causes obstruction of the blood flow with subsequent myocardial ischemia. Its pathogenesis is not understood to its full extent, but hypotheses lie mainly on spontaneous intramural hemorrhage formation in the arterial wall or coronary flow mediated expansion after an intimal tear. This disease is of growing interest, as it has been more diagnosed with intravascular imaging methods [25,26,27]. 

Eleid et al. showed that coronary artery tortuosity may be a marker for or a potential mechanism for SCAD [28]. 

Other predisposing factors would include fibromuscular dysplasia (FMD), postpartum status, multiparity (≥4 births), connective tissue disorders, systemic inflammatory conditions, and hormonal therapy [29,30,31].

SCAD most commonly occurs in patients with few or no traditional cardiovascular risk factors [32,33]. 

Recent studies have shown that SCAD occurs overwhelmingly in women [34,35] and is the most common cause of pregnancy-associated AMI [36,37]. A recent analysis of a US administrative database found a prevalence of 1.81 SCAD events per 100,000 pregnancies during pregnancy or in the 6-week postpartum period [38]. The left main or left anterior descending artery has been described as the most commonly affected [37,39]. The pathophysiological mechanism involved in this phenomenon is not fully understood, but it is possibly associated with numerous conditions such as hormonal changes of pregnancy that may lead to alterations in the architecture of the arterial wall [40,41]. 

In addition to hormonal influences, other situations are associated with SCAD such as underlying arteriopathies, genetic factors, inherited or acquired arteriopathies, or systemic inflammatory diseases, often compounded by environmental precipitants or stressors [33]. 

SCAD can be classified based on angiographic appearance into four types (Figure 2) [24,26]. Type 1 angiographic SCAD describes those with evident arterial wall stain with multiple radiolucent lumens. Type 2 A refers to a segment with diffuse narrowing (typically >20 mm) with “normal” segment proximally and distally. Type 2 B refers to diffuse narrowing extending to the distal end of the vessel. Type 3 refers to short segment of stenosis (<20 mm in length) that mimics atherosclerosis. Type 4 is characterized by dissection leading to an abrupt total occlusion, usually of a distal coronary segment.

Observational studies have shown that the majority of patients (70–97%) have angiographic healing weeks to months after a conservatively managed index episode [35,42,43]. 

The time course of healing remains uncertain, but it can be detected within days [44]. 

The diagnosis of SCAD is usually possible with coronary angiography alone, but intravascular imaging such as IVUS or OCT is paramount for detecting more challenging SCAD cases (Figure 3). However, while OCT images in SCAD are characteristic, IVUS images require closer scrutiny to discriminate between plaque disruption and SCAD, given the lower spatial resolution of IVUS [45]. However, OCT could further aggravate the dissection or exacerbate a new intimate tear due to contrast injection. In addition, the more recent high-definition IVUS (HD IVUS) has better spatial resolution, which helps with the diagnosis of SCAD [46].

Coronary computed tomography angiography (CCTA) has been utilized both in the initial diagnosis of SCAD and to assess healing. However, CCTA diagnostic criteria for SCAD need further refinement. During the acute SCAD episode, dissection planes are infrequently identified (<15%) by CCTA; abrupt luminal changes and sleeve-like hematomas within the coronary artery wall are more often observed [47,48]. Limitations of CCTA include low spatial resolution for small vessels and with a diameter < 2.5 mm and for the middle and distal portions of the coronary arteries, motion artifact, and unknown sensitivity and specificity [45]. Therefore, CCTA can have its place for proximal SCAD but very few for mid and distal SCAD which is the preferred location for SCAD.

### 6.2. Coronary Artery Spasm (CAS)

Vasospastic angina (VSA) was first described by Prinzmetal [49,50]. Recent studies have shown a high prevalence of epicardial (26–37%) and microvascular spasm (33–34%) in patients undergoing coronary angiography [51,52]. 

Therefore, CAS can occur at the level of the epicardial arteries as well as in the coronary microcirculation. Current standardized diagnostic criteria for microvascular spasm include reproduction of the patient’s angina symptoms and ischemic ECG changes in the absence of epicardial spasm during intracoronary spasm provocation testing using, for example, acetylcholine [53]. Of note, it is important to mention that it is difficult to identify the mechanism of microvascular dysfunction that triggers microvascular angina. Therefore, it is essential to distinguish between an impaired microcirculatory vasodilatory capacity, which can be diagnosed by measuring coronary flow reserve or microvascular resistance, and microvascular spasm determined by intracoronary acetylcholine administration.

CAS usually occurs at a localized segment of an epicardial artery (Figure 4), but sometimes involves two or more segments of the same (multifocal spasm) or of different (multivessel spasm) coronary arteries.

Of interest, myocardial bridging, per se, is unlikely to cause MINOCA [54]. However, it can predispose the affected artery to spasm [55,56].

Proposed mechanisms to constitute the substrate for CAS susceptibility include: (1) endothelial dysfunction and (2) primary hyperreactivity of vascular smooth muscle cells [57].

The endothelium plays a crucial role in the physiological regulation of coronary vascular tone, mainly through the release of vasodilating substances, the most important of which is nitric oxide (NO). Therefore, significant endothelial damage can impair vasodilation, favoring CAS in response to vasoconstrictor stimuli [58]. Various vasoactive stimuli (e.g., acetylcholine, serotonin, histamine) cause vasodilation by inducing nitric oxide (NO) release from the endothelium, but at the same time can cause vasoconstriction by direct stimulation of vascular smooth muscle cells. Thus, in the presence of endothelial dysfunction, its release into the vessel wall can lead to vasoconstriction or coronary spasm.

Additionally, there is consistent evidence to suggest that, in patients with variant angina, a primary nonspecific hyperreactivity of vascular smooth muscle cells in the coronary artery wall is the main abnormality responsible for coronary spasm. The pathogenic role of local hyperreactivity of vascular smooth muscle cells is suggested by the observation that vasoconstrictor stimuli that induce spasm in localized coronary segments of patients with variant angina are unable to induce spasm in other coronary segments of the same patients [59] and in patients with other forms of angina (in particular, stable angina) [60,61]. 

Furthermore, in patients with variant angina, CAS can be triggered by various stimuli that act through different receptors and cellular mechanisms [62,63], suggesting an intracellular and post-receptor location of the alteration responsible for the hyperreactivity.

Patients with VSA typically have recurrent episodes of angina with no clear relation to exercise and metabolic demand (Figure 5). Classically, these episodes are accompanied by ST deviations on ECG and prompt relief with nitrates. CAS attacks can occur both in patients with and without obstructive atherosclerosis [64].

Features that are specific to VSA (versus classical angina) [64]:Angina occurs predominantly at rest and may occur more frequently from midnight to early morning;Effort and exercise tolerance are usually preserved;Hyperventilation can precipitate VSA [65];Episodes appear in “clusters;”VSA often has a more rapid response to sublingual nitroglycerin.

The diagnosis of VSA typically requires the documentation of CAS. Coronary angiography may be normal due to the self-limiting nature of the episodes. Therefore, the diagnosis of VSA may require provocative stimulus using acetylcholine, ergonovine, or methylergonovine (Figure 6). A positive provocative test for CAS must induce all of the following in response to the provocative stimulus: (1) reproduction of the usual chest pain, (2) ischemic ECG changes, and (3) >90% vasoconstriction on angiography [64]. 

Although accurate [66], provocative tests for CAS are associated with risks. The risk of death is low, but the incidence of cardiac arrhythmias is relatively high (6.8%) [67]. A contemporary study analyzed 921 patients undergoing intracoronary acetylcholine testing, and no deaths or serious complications were reported [68]. Also, only 1% of patients had any kind of complications, namely non-sustained ventricular tachycardia, fast paroxysmal atrial fibrillation, bradycardia with hypotension and catheter-induced vasospasm. Although complications related to provocative tests may exist, none of them are associated with increased morbidity and mortality.

Montone et al. evaluated 80 patients suspected of a vasomotor etiology with acetylcholine or ergonovine intracoronary provocative tests [69]. Out of 37 positive tests, only two patients (5.4%) presented with arrhythmic complications (self-limited bradycardia). The authors reported a rate of complications comparable to that of spontaneous vasomotor angina episodes. Taking a closer look at the Montone series, the rate of test positivity was significant, which highlights the importance of this etiology in the pathogenesis of MINOCA: 37 out of 80 selected patients had a positive test; 24 had epicardial spasm and 13 had microvascular spasm.

The significance of this vasomotor change is likely similar to that found in ruptured plaques and evidence of a mechanism by which MINOCA is plausible, but a definitive causal association is probably not possible in all cases.

### 6.3. Coronary Artery Embolism/Thrombosis

Coronary artery embolism (CE) is an uncommon cause of AMI and the precise diagnosis remains challenging for the physician. A recent retrospective analysis suggested that up to 3% of AMI might result from CE [70]. 

The National Cerebral and Cardiovascular Center group proposed diagnostic criteria to define CE (Table 2) [70]: 

CE is divided into four groups: (1) direct; (2) paradoxical; (3) iatrogenic; and (4) hypercoagulable disorders, with some overlap among the categories. 

Direct coronary emboli commonly originate from the left atrial appendage, left ventricle, the aortic or mitral valves, or the proximal coronary artery. Embolic tissue may be thrombus, valvular material, or even neoplasm. 

Paradoxical emboli pass through a patent foramen ovale (PFO), atrial septal defect, or pulmonary arteriovenous malformations from the venous circulation into the systemic circulation. Most commonly the origin is from a deep vein (deep venous thrombosis). 

Iatrogenic emboli may occur following interventional procedures, usually valve replacements and coronary intervention.

Hypercoagulable disorders that result in coronary thrombosis can be divided into inherited and acquired causes.

Inherited thrombophilia includes factor V Leiden, elevated factor VIII/von Willebrand factor, activated protein C resistance, protein C or S deficiency, prothrombin 20120A, and antithrombin deficiency. Acquired hypercoagulable states include thrombotic thrombocytopenic purpura (TTP), the autoimmune disorder antiphospholipid syndrome, heparin-induced thrombocytopenia (HIT), and myeloproliferative neoplasms.

The diagnosis of CE is based on the clinical presentation and the presence of risk factors. IVUS or OCT can help differentiate spontaneous coronary thrombosis or embolization from other MINOCA etiologies such as plaque rupture. Echocardiography, transesophageal echocardiography, and microbubble studies are helpful in finding the source of emboli. Thrombophilia can be investigated through specific tests.

### 6.4. Coronary Microvascular Dysfunction (CMD)

The term microvascular angina was initially proposed by Cannon and Epstein in 1988 to identify patients with myocardial ischemia triggered not by obstructive CAD, but by functional microvascular abnormalities [71]. Coronary microvascular dysfunction is also commonly referred to as syndrome X.

Microvascular circulation (vessels <0.5 mm in diameter) is not visualized on coronary angiography and represents approximately 70% of coronary resistance in the absence of obstructive CAD (Figure 7). The dysfunction affects only these vessels, and it is characterized by reduced coronary flow reserve (CFR) [72]. 

CFR is an invasive method that allows an integrated measurement of flow through the large epicardial arteries and coronary microcirculation, but once severe obstructive disease of the epicardial arteries is ruled out, reduced CFR is a marker of CMD. CFR is the ratio of hyperemic blood flow divided by resting blood flow and can be calculated using thermodilution or Doppler flow velocity. Overall, the prognostic value of CFR used a cutoff value < 2.0 [73]. The index of microcirculatory resistance (IMR) is calculated as the product of distal coronary pressure at maximal hyperemia multiplied by the hyperemic mean transit time. IMR ≥ 25 is representative of microvascular dysfunction [74,75,76]. Flow-limiting obstructive coronary artery disease can be assessed using Fractional Flow Reserve (FFR), which is the ratio of mean distal coronary pressure to mean aortic pressure at maximal hyperemia (abnormal FFR is defined as ≤0.80) [73]. FFR values > 0.8, CFR ≥ 2.0, and IMR < 25 represent absence of CMD and after vasoactive stimuli with acetylcholine with absence or reduction of coronary diameter < 90%, without angina and lack of ischemic ECG changes it is interpreted as pain non-cardiac and the opposite changes in the test allow the diagnosis of VSA. The FFR values > 0.8, CFR < 2.0, and IMR ≥ 25 represent the presence of CMD and after a vasoactive stimuli with acetylcholine with absence or reduction of coronary diameter < 90%, without angina and lack of ischemic ECG changes it is interpreted as microvascular angina and the opposite test result allows diagnosis of microvascular angina and VSA.

Recently, Rahman et al. described two endotypes of CMD: structural and functional [77]. In structural CMD patients have endothelial dysfunction, which leads to diminished peak resting coronary blood flow augmentation and increased demand during exercise the functional CMD is related to inefficient cardiac-coronary coupling during peak exercise and during rest leads to higher myocardial oxygen demand in the setting of exhausted vasodilatory reserve [78].

The Women’s Ischemia Syndrome Evaluation (WISE) study showed that the prevalence of microvascular dysfunction and nonobstructive CAD is high and is associated with relatively poor prognosis compared with women without evidence of microvascular dysfunction and nonobstructive CAD [79,80].

Recently, the COVADIS group proposed diagnostic criteria to define microvascular angina (Table 3) [53]. 

The mechanism of microvascular dysfunction may be endothelium-dependent or endothelium-independent [81]. Endothelium-dependent dysfunction is a consequence of an imbalance between relaxing factors, such as NO, and constricting factors, such as endothelin. Endothelium-independent dysfunction is based on myocyte tone [81]. In addition, enhanced coronary vasoconstrictive reactivity and increased coronary microvascular resistance secondary to structural factors (e.g., luminal narrowing, vascular remodeling, vascular rarefaction, and extramural compression) are also involved in microvascular dysfunction [82].

Assessment of microvascular dysfunction includes invasive methods such as CFR, index of microvascular resistance (IMR), and absolute coronary blood flow measured, and non-invasive methods such as positron emission tomography (PET), CMR, and Doppler echocardiography.

The concept behind CFR lies in the prerogative that the vessels in the coronary tree have the potential to heighten their flow in response to vasodilator stimuli. This potential is the so-called flow reserve. In ischemic territories, endogenous mechanisms will lead to a basally more dilated coronary bed, and therefore with less flow reserve [83]. Measuring CFR attempts to evaluate the potential to increase blood flow in response to specific stimuli. Both epicardial obstructions and microvascular dysfunction may lead to lower CFR, thus CFR is only applicable to diagnose the latter in the absence of significant epicardial obstruction. CFR is calculated using thermodilution as resting mean transit time divided by hyperemic mean transit time. However, CFR has some limitations such as: (1) low specificity for microvascular dysfunction; (2) it does not have a clearly defined normality value; (3) is affected by hemodynamic variables at rest. In the Women’s Ischemia Syndrome Evaluation (WISE) study, a total of 47% of women had diminished coronary flow velocity reserve (CFVR), suggestive of microvascular dysfunction [84]. 

The index of microcirculatory resistance (IMR) was firstly developed by Fearon et al. and is calculated from estimates of maximal distal coronary flow during hyperemia and pressure [85]. Ng et al. showed that IMR is superior to CRF for assessing the coronary microcirculation by virtue of being more reproducible and less hemodynamically dependent than CFR [86] since it is not dependent on resting values. Moreover, IMR is not affected by epicardial stenosis severity [87]. Other indices can be used to assess CMD, such as hyperemic microvascular resistance [88], resistive reserve ratio [89], and microvascular resistance reserve (MRR) [90].

Recently, the novel technique to quantify absolute coronary flow and resistance through intracoronary continuous thermodilution has been developed. Morris et al. has demonstrated that this new method provides a comprehensive coronary physiological assessment of flow, pressure and resistance, across the entire coronary circulation, without the need for additional hardware, catheters, wires, or infusions [91].

However, the body of evidence concerning coronary flow and flow reserve measurement among the MINOCA population is currently limited. Similarly, the role and the clinical implications of continuous thermodilution-derived indexes within MINOCA patients are not yet established [92].

### 6.5. Supply–Demand Mismatch

Stable CAD typically does not cause myocardial necrosis because the obstruction grade is fixed. However, in states of intense demand, these obstructions may lead to critical hypoperfusion, with the development of AMI and necrosis. In extreme demand scenarios, this can occur even in the absence of obstructive coronary lesions (Figure 8). 

Approximately 50% of patients with type II AMI do not have significant coronary artery disease, and they can be classified as MINOCA [93]. 

The common causes of demand and supply mismatch are hypotension, tachyarrhythmia, and hypoxia [76]. 

Along with the widespread introduction of high-sensitivity troponin assays, the detection of abnormal levels of these cardiac biomarkers in hospitalized patients has been frequent [94].

The Fourth Universal Definition of Myocardial Infarction delineates the principles by which the clinician may establish the differential diagnosis between AMI and myocardial injury (Figure 9) [95]. Any rise and/or fall in troponin level with at least one result over the 99th percentile is a myocardial injury. On the other hand, the alteration of cardiac biomarkers associated with ischemic findings (ECG, symptoms and imaging exams) defines the diagnosis of AMI. The diagnosis of a type 2 AMI, as opposed to myocardial injury, requires ischemic symptoms or signs and a rise or fall in troponin levels. The presence of CAD is not necessary for the diagnosis. 

Interestingly, several mechanisms not involving intracoronary thrombus are also categorized as type 2 AMI, such as SCAD, vasospasm, and microvascular dysfunction. Nevertheless, the classical clinical picture of an acute myocardial injury or a type 2 AMI is that of a patient in sepsis, or with severe tachycardia and/or hypertension, evolving with elevated biomarkers.

## 7. Non-Schemic Differential Diagnosis

### 7.1. Takotsubo Cardiomyopathy

Takotsubo cardiomyopathy (TCM) is also known as stress cardiomyopathy or ‘broken heart syndrome’ and often mistaken with MI. TCM was first described in 1990 by Sato et al. [96] and is characterized by transient left ventricular dysfunction, with clinical, electrocardiographic, and laboratory characteristics similar to those of ACS. The syndrome occurs predominantly in postmenopausal women [97] after an episode of physical or psychological stress. It is estimated that approximately 1–2% of patients with a primary diagnosis of ACS have TCM [98]. Its pathophysiology is not yet fully understood, but excess catecholamines have been postulated to be central to the pathogenesis of TCM.

Since patients with TCM commonly present with chest pain, electrocardiographic changes, and high troponin levels, the differential diagnosis with myocarditis and ACS remains challenging, particularly when concurrent coronary artery disease is present (15% in the International Takotsubo Registry) (Table 4) [99]. Therefore, early coronary angiography remains necessary to rule out an ACS.

Although the majority of patients with TCM recover cardiac function spontaneously within several weeks [100], the in-hospital mortality rate is relatively high [101]. 

Scally et al. analyzed the chronic phase of TCM in a cohort of 37 patients [102]. Despite the normal left ventricular ejection fraction and serum biomarkers, at rest, 88% of TCM patients had persisting symptoms with heart failure, limitations on exercise cardiopulmonary testing, impaired cardiac strain and deformation compatible indices on transthoracic echocardiography, and increased native T1 mapping values and increased impaired cardiac energetic status on CMR.

### 7.2. Myocarditis

Myocarditis is an inflammatory disease of the heart that may occur because of infections, immune system activation, or exposure to drugs [103]. Myocarditis has different causes, with infectious being the most common. Acute myocarditis may present with symptoms similar to ACS, with chest pain, electrocardiographic changes, and elevation of myocardial necrosis markers.

The publication of the Fourth Universal Definition of Myocardial Infarction established that the term MINOCA would encompass only ischemic mechanisms of myocardial injury and thus both myocarditis and TCM were excluded from this definition [95].

In practice, the differential diagnosis between MINOCA and non-ischemic causes of myocardial injury is always a challenge. Study with CMR in patients with an initial diagnosis of MINOCA showed myocarditis as the main cause (33%), with the true ischemic mechanism responsible for only 19% of cases [104]. 

CMR can be useful in the differential between myocarditis, MINOCA and TCM, but endomyocardial biopsy remains the gold standard for the diagnosis of myocarditis, although is not necessary in most cases. Common CMR findings are mesocardial and epicardial edema, with a speckled pattern, in addition to late mesocardial and epicardial enhancement [105]. 

## 8. Clinical Investigation

The physician must delineate an investigation strategy to reach the correct diagnosis. There is no single strategy with proven cost-effectiveness superiority, and therefore the clinician should tailor the investigation according to the pretest probabilities of an individual patient. Facing a patient with MINOCA, the tests should be focused on answering two questions (Figure 10):Is this really MINOCA?Which exact mechanism caused the myocardial injury?

The first step when facing a patient with an AMI with nonobstructive coronary arteries is to guarantee that the angiogram is truly free from obstructive lesions. A thorough review of the coronary angiography is necessary, as certain lesions may be overlooked, especially distal occlusions and SCAD.

Even after coronary angiography without obstructive lesions, it is still possible that the event is not secondary to an AMI. As seen previously, there are nonischemic diagnoses that can mimic AMI, such as myocarditis and TCM. The best test for this purpose is the CMR.

The second question may be answered with probability driven investigations. As discussed previously, plate disruption can be diagnosed with intravascular imaging, as well as SCAD. CAS may be seen spontaneously during the catheterization or may be provoked with the tests described previously. CMD can be diagnosed invasively or noninvasively, primarily with the aid of CFR. Thrombophilia may be diagnosed with blood tests. Finally, supply–demand mismatch can be inferred due to the clinical scenario.

### 8.1. The Role of the CRM

Recent studies have suggested that an initial workup consisting of multimodality assessment, including CMR can yield the diagnosis in most patients with MINOCA [106,107]. CMR is able to identify if the lesion pattern is compatible with an ischemic or nonischemic mechanism (Figure 11) and enables the direct visualization and quantification of microvascular obstruction (MVO), which reflects myocardial damage resulting from microvascular dysfunction [108,109]. The diagnosis of MVO by CMR is performed by first pass perfusion (FPP) and late gadolinium enhancement (LGE) techniques. The assessment of MVO is based on the distribution characteristics of gadolinium in different tissues, such as normal, recent infarction and fibrotic myocardium. Gadolinium is distributed in extracellular spaces. Normal myocardium has less extracellular space than infarction and fibrosis. Thus, infarcted and fibrotic myocardial regions appear clear due to the greater distribution of contrast in these regions. No-reflow regions or, more precisely, MVO regions appear as dark areas within the infarcted area (Figure 11 and Figure 12). The optimal timing for performing CMR is probably between 24 h and 7 days, to avoid false-negative results [107,110]. 

LGE is useful to identify the pattern of myocardial injury (Figure 13). The pattern encountered in ischemic conditions is subendocardial or transmural. Mesocardial and subepicardial patterns suggest nonischemic causes, such as myocarditis. The absence of LGE does not exclude a myocardial infarction, but its presence with the described ischemic patterns corroborates the diagnosis of MINOCA and reinforces the need for additional investigations.

In addition, T2 and T1 sequences allow the search for edema and extracellular volume, which can be markers of myocardial injury. Both TCM and myocarditis can be confirmed due to the presence of edema in the CMR [111,112]. Specifically for myocarditis, the International Consensus Group on CMR Diagnosis of Myocarditis published detailed recommendations on the analysis of appropriate CMR techniques for diagnosis of myocarditis (Lake Louise criteria) (Table 5) [113]. 

Furthermore, it is important to recognize that the diagnostic accuracy of CMR for myocarditis varies according to the clinical presentation of the disease. The sensitivity of T2-weighted imaging in the chronic stages of myocardial inflammation has been questioned [110,112,113,114] whereas native T1 mapping does not appear to be able to discriminate between acute and chronic disease [115]. 

However, the Stockholm Myocardial Infarction with Normal Coronaries 2 (SMINC-2) study showed that CMR performed early (median of 3 days) as a diagnostic tool in the investigation of patients with MINOCA is highly effective (77% of patients with MINOCA received a diagnosis) [116]. Additionally, Pathik et al. also showed that CABG was able to establish the diagnosis in 87% of MINOCA patients with a median time between CMR and admission of 6 days [117]. 

### 8.2. The Role of Intravascular Imaging 

Intravascular imaging includes IVUS and OCT. Both are performed during coronary catheterization and can clarify the vascular anatomy, detecting plate disruption and SCAD. OCT has a better resolution than IVUS, allowing better tissue characterization, detection of thrombus, and identification of culprit lesions in patients with suspected ACS [118]. However, aorto-ostial lesions are not evaluable by OCT due the difficulty obtaining adequate proximal blood flush [119]. Furthermore, it is known that OCT requires more contrast for acquisition of images and therefore is not the preferable technique in patients with chronic kidney disease (CKD) [119]. IVUS is preferable in patients with CKD because of the lower volume of contrast needed for coronary angiography and OCT [120,121]. 

Studies have suggested a very high diagnostic yield with the combination of both intravascular imaging and CMR (84–100%) [17,122]. Probably, intravascular imaging has the highest sensitivity when performed early in the course of MINOCA (i.e., by the time of the first angiogram with nonobstructive coronary arteries) [107]. 

Both IVUS and OCT are safe procedures. A single-center study evaluated adverse events in patients undergoing invasive imaging during coronary catheterization [123]. Invasive imaging-related complications were rare, did not differ between the two imaging methods (0.5–0.6%). No major adverse events, prolongation of hospital stay, or permanent patient harm was observed. However, OCT has only been recommended when the diagnosis of SCAD is uncertain, since the introduction of the guidewire into the lumen can end up in the false lumen inducing SCAD extension.

## 9. Management

Optimal management depends upon correct diagnosis of the underlying MINOCA mechanism (Figure 14). 

Analysis of the SWEDEHEART registry with 9136 MINOCA patients showed that most patients received at least one of the following medications: statins, angiotensin-converting enzyme inhibitors (ACEi) or angiotensin receptor blockers (ARB), beta-blockers (BB), and dual antiplatelet therapy (DAPT) [124]. The mean follow-up was 4.1 years, with a 23.9% rate of major cardiac events (MACE). Using the propensity score, the study reported favorable hazard ratios for patients receiving ACEi/ARB (HR 0.82, 95% CI 0.73–0.93) or statins (HR 0.77, 95% CI 0.68–0.87). Currently, the MINOCA-BAT study is prospectively studying the effects of beta-blockage and ACEi/ARBs in patients with MINOCA [125]. 

Specific management for identified etiologies will be discussed here. There will not always be a clearly identified diagnosis, and management must be based on the probabilities and risk-benefit balance of each treatment.

### 9.1. Plaque Disruption

Since the mechanism of MINOCA caused by plaque disruption is the same as in AMI-CAD, the usual therapy is similar to that of patients with ACS.

DAPT should be initiated and follow the same principles as delineated by the guidelines [126]. 

The positive effects of statins and ACEi/ARB found in the SWEDEHEART registry probably apply to these patients [126]. Recent meta-analysis has shown that statins significantly reduce the rate of MACE and mortality in patients with MINOCA [127]. 

No studies to date have evaluated the effects of BBs in patients with MINOCA.

Care should be taken, however, not to assume that there is robust evidence of significant and relevant benefit of any specific therapy. As for all that involves MINOCA, science is still progressing and *primum non nocere* must always be considered.

In patients with AMI-CAD with a patent vessel, a routine invasive strategy with revascularization when appropriate is well-established. Several trials have shown the benefits of the invasive strategy derived primarily from reduced AMI and refractory angina [128,129]. These studies, however, performed coronary revascularization “if appropriate”, which precludes generalization to the MINOCA population. To the best of our knowledge mechanisms, there is no evidence to support routine percutaneous coronary intervention (PCI) in these patients, even with atherosclerotic as the cause. 

### 9.2. SCAD

The principle for the management of SCAD is based on the understanding that this pathology consists of an intramural hematoma that has a high rate of reabsorption in the medium term of follow-up.

The treatment of SCAD is controversial. There is a lack of randomized controlled trials that have evaluated treatment strategies in this setting. However, as 70–97% of patients managed conservatively show angiographic healing of SCAD lesions [43,44], conservative treatment is recommended in most cases.

The combination of aspirin and BB has been recommended for the long-term medical treatment of SCAD, although there are no randomized clinical trials (RCTs) on the issue. However, an observational study showed that BB has been associated with a lower risk of recurrent SCAD (HR 0.36, *p* = 0.004) [31], probably because BB is able to reduce shear stress on the vessel wall and minimize risk of propagation [130]. Currently, the Beta-Blockers and Antiplatelet Agents in Patients with Spontaneous Coronary Artery Dissection (BA-SCAD) trial is ongoing, which aims to assess the efficacy of medical therapy in SCAD patients [131]. 

Although DAPT should be administered to patients who undergo PCI, DAPT for acute management of SCAD patients not treated with stents is of uncertain benefit. An expert consensus states that a course of DAPT may be considered during the acute phase of SCAD [132] and for up to 12 months [33]. On the other hand, the multicenter DISCO (DIssezioni Spontanee COronariche) registry showed that conservatively managed SCAD patients who received DAPT had a higher rate of adverse cardiovascular events at 1 year compared with those who received a single platelet regimen [133]. 

ACEi or ARB has not been studied in SCAD patients. Currently, the SAFER-SCAD (Statin and Angiotensin-converting Enzyme Inhibitor on Symptoms in Patients With SCAD; NCT02008786) trial is ongoing and aims to assess whether these medications reduce chest pain and improve small vessel function in SCAD patients.

PCI in SCAD patients has been associated with more adverse events, propagation of the hematoma, and worse final results. A retrospective study found PCI failure rate of 53% without protection against target vessel revascularization or recurrent SCAD [42]. Therefore, coronary revascularization is recommended only for patients at high risk due to involvement of multivessel severe proximal dissections or of the left main artery or the ostial left anterior descending artery, hemodynamic instability, or refractory arrhythmia [24]. 

Although the experience with cutting balloon (CB) angioplasty for SCAD is limited, recent studies have compared CB with PCI for SCAD [134,135]. CB angioplasty can fenestrate the false lumen to allow communication and back-bleed of intramural hematoma into the true lumen.

Future pregnant women with previous SCAD should be counseled regarding the risk of recurrence. Previous reports have shown that most pregnancies are uneventful, although serious adverse events have been reported [136]. The multidisciplinary team should participate in counseling and monitoring of future high-risk pregnant women [137]. If the patient decides on contraception, the preferred contraceptive options are partner sterilization, tubal ligation, or intrauterine devices [138]. 

### 9.3. Coronary Artery Spasm (CAS)

Management of CAS is based on (1) identifying and avoiding precipitating factors such as smoking, drugs, and excessive fatigue and mental stress, (2) physical activity [139], (3) control of risk factors such as hypertension, diabetes, and hypercholesterolemia, and (4) pharmacological therapy.

Calcium channel blockers (CCBs) represent for patients with coronary spasm the mainstay of therapy [140,141]. In refractory cases, a combination of two CCBs (dihydropyridine and non-dihydropyridine) may be required, although there is a paucity of RCTs [142]. Short-acting formulations of nifedipine should not be used because they are more likely to result in rebound symptoms.

Nitrates are effective in reducing anginal episodes [141,143]. Long-acting nitrates can be proposed in addition to CCBs to improve the efficacy of the treatment. However, there is a relatively high occurrence of nitrate tolerance, and long-acting nitrates may trigger endothelial dysfunction [144]. 

Beta-blockers should be avoided in patients with CAS as they can exacerbate and prolong vasospasm [145,146]. 

Studies have shown benefits of statins in preventing CAS [147]. However, the impact of statins in reducing cardiovascular events in patients with VSA is still controversial [148,149,150]. 

Other pharmacological therapies with less robust evidence may be used to suppress inducible CAS such as nicorandil [151], magnesium [152], estrogen replacement therapy in postmenopausal women [153], and vitamin C therapy [154]. 

Finally, aspirin should be avoided at high doses, as it is an inhibitor of prostacyclin production at high doses [155] and used with caution in low doses [156]. 

### 9.4. Coronary Artery Embolism/Thrombosis 

Treatment options are based on the specific etiology and precipitating factors of CE. In patients with high thrombotic burden, aspiration thrombectomy may be considered [157]. Those with atrial fibrillation, left ventricular or left atrial appendage thrombus should be treated with long-term oral anticoagulation, regardless of CHADS2 or CHADS2-VASc score [158]. Anticoagulation should be achieved with warfarin or a novel oral anticoagulant [159]. In cases of suspected paradoxical embolism, definitive surgical closure of a patent foramen ovale or atrial septal defect may help prevent recurrent episodes of AMI. In patients with clotting disorders, a hematologist should be consulted.

### 9.5. Coronary Microvascular Dysfunction

Currently, there is a lack of evidence of validated treatment that has proven to be effective in patients with MVA. A recent systematic review that evaluated different treatment options did not find sufficient data to support therapies for MVA [160]. However, the European Society of Cardiology guideline recommends beta-blockers, ACEi, and statins for patients with abnormal CFR < 2.0 or IMR ≥ 25 units along with lifestyle changes and weight loss [161]. 

### 9.6. Supply–Demand Mismatch

The management of patients with AMI resulting from a supply–demand mismatch is the treatment of the underlying disease. Since, by definition, there are no obstructive coronary lesions, additional therapy is usually not necessary.

## 10. Future Directions

A significant knowledge gap still persists regarding diagnostic methods and management of patients with MINOCA. Therefore, future research should focus on these two domains:The identification of the MINOCA mechanism has important implications for management and secondary prevention and a multimodality imaging approach should be employed for a prompt recognition of MINOCA.To evaluate the best strategy for the management and follow-up of patients with MINOCA in order to prevent cardiovascular events.

## 11. Conclusions

MINOCA is a heterogeneous working diagnosis requiring further investigation, including the use of intracoronary imaging (IVUS or OCT) and CMR with LGE. MINOCA is not benign and has comparable outcomes with AMI-CAD, according to the findings of the VIRGO study [2]. Treatment of the underlying cause is paramount although often empirical, but once the cause is established, targeted evidence-based treatment should be instituted.

## Figures and Tables

**Figure 1 jcm-11-05497-f001:**
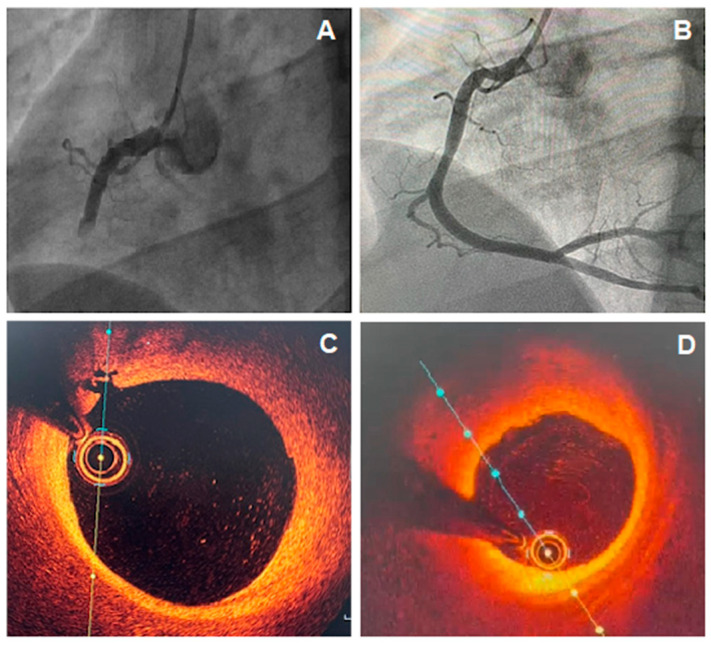
(**A**) Coronary angiography showing total occlusion of the right coronary artery (RCA); (**B**) Coronary angiography of RCA, 5 days after recanalization and intracoronary thrombolytic administration, showing non-obstructive atherosclerotic plaque; (**C**,**D**) optical coherence tomography (OCT) images showing a lipid plaque with signs of inflammation and intimal rupture.

**Figure 2 jcm-11-05497-f002:**
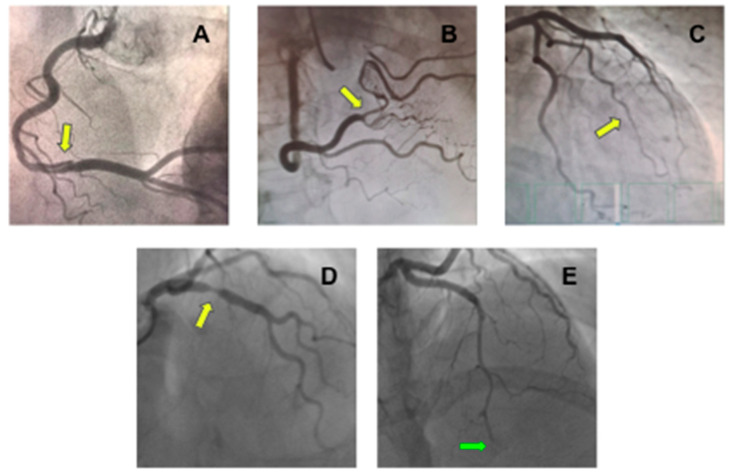
Coronary angiography showing different types of spontaneous coronary artery dissection (SCAD). (**A**) An image of double lumen at right coronary artery (yellow arrow) compatible with type 1 SCAD; (**B**) Stenosis at right coronary artery with distal vessel caliber normalization (yellow arrow) compatible with type 2A SCAD; (**C**) Long stenosis from mid-to-distal first marginal branch of the left circumflex coronary (yellow arrow) compatible with type 2B SCAD; (**D**) Focal stenosis at right coronary artery (yellow arrow) compatible at type 3 SCAD; and (**E**) Stenosis from mid-to-distal left anterior descending coronary artery (green arrow) with an abrupt total occlusion (green arrow) compatible with type 4 SCAD.

**Figure 3 jcm-11-05497-f003:**
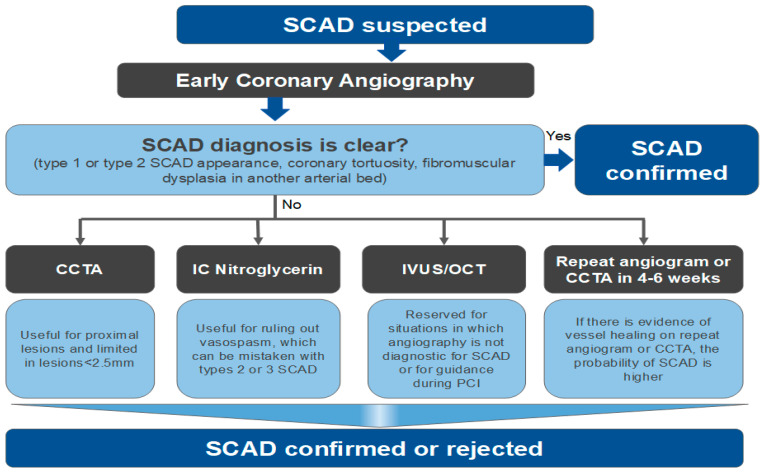
Flowchart for the diagnosis of spontaneous coronary artery dissection (SCAD). CCTA: Coronary Computed Tomography Angiography; IVUS: Intravascular ultrasound; OCT: Optical coherence tomography; IC: intracoronary; PCI: percutaneous coronary intervention.

**Figure 4 jcm-11-05497-f004:**
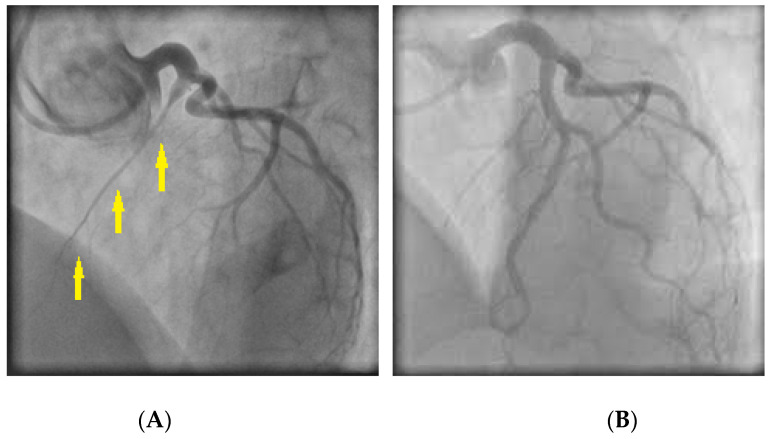
(**A**) Coronary angiography revealing spasm in the right coronary artery (arrows); (**B**) Response of the stenotic region to intracoronary nitroglycerin administration.

**Figure 5 jcm-11-05497-f005:**
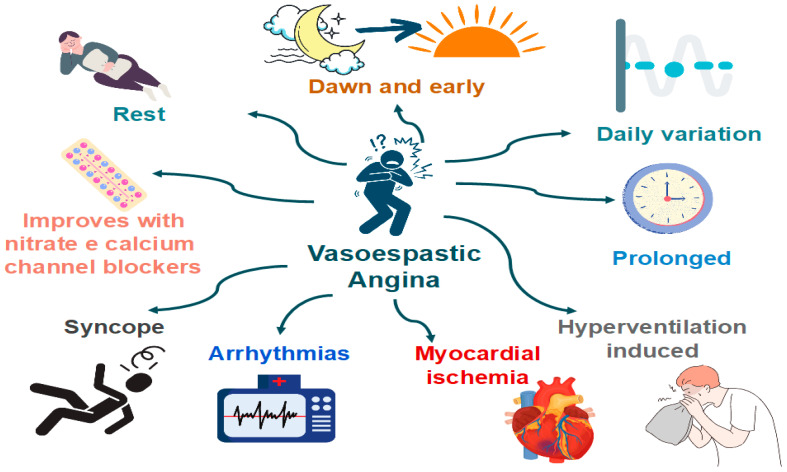
Clinical manifestations of coronary artery spasm (CAS).

**Figure 6 jcm-11-05497-f006:**
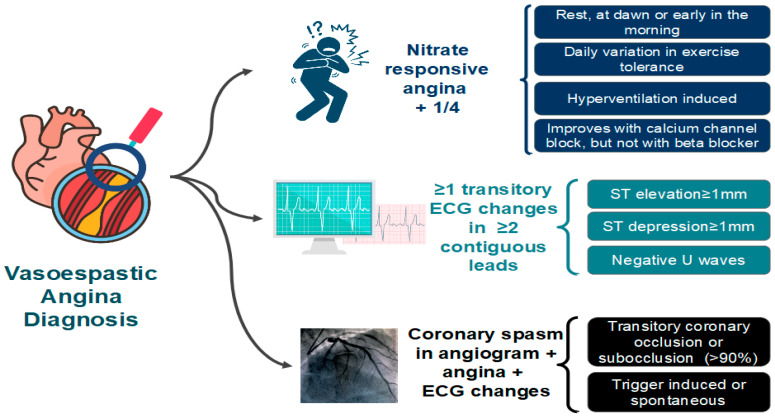
Diagnostic criteria for vasospastic angina (VSA) proposed by the Coronary Vasomotion Disorders International Study Group (COVADIS). ECG: electrocardiogram [64].

**Figure 7 jcm-11-05497-f007:**
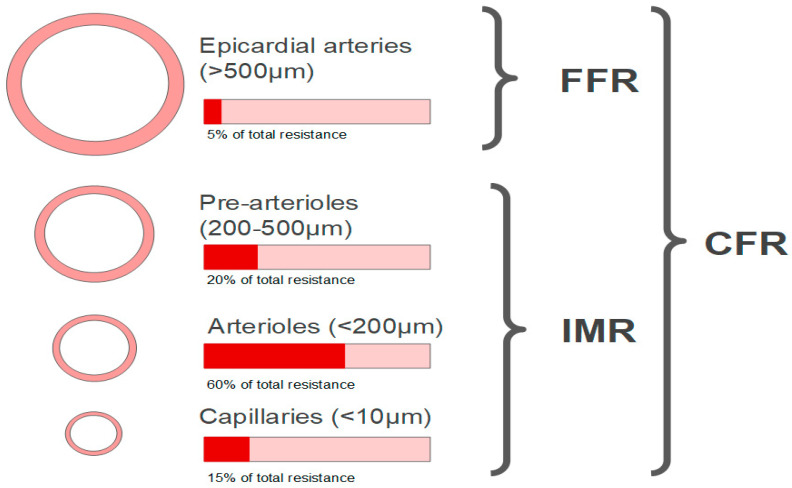
Anatomy of the coronary arterial system and invasive diagnostic modalities to assess coronary microvascular function. FFR: fractional flow reserve; IMR: index of microvascular resistance; CFR: coronary flow reserve.

**Figure 8 jcm-11-05497-f008:**
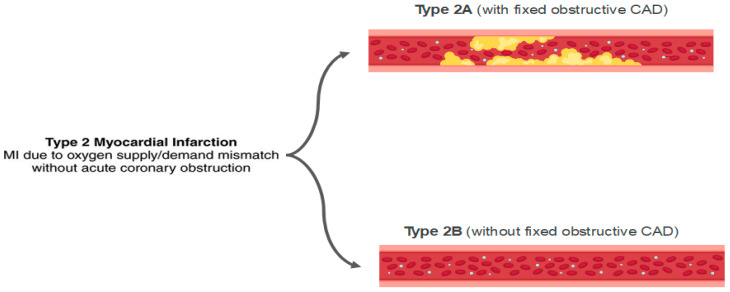
Classification of type 2 acute myocardial infarction (AMI). MI: myocardial infarction; CAD: coronary artery disease.

**Figure 9 jcm-11-05497-f009:**
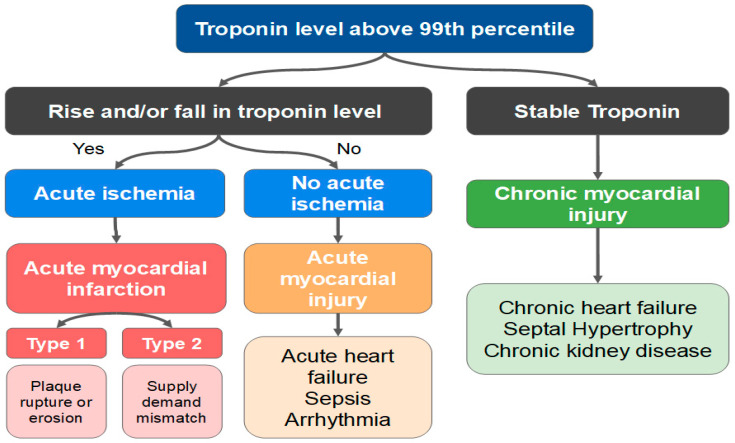
Differences between acute myocardial infarction (AMI) and acute/chronic myocardial injury.

**Figure 10 jcm-11-05497-f010:**
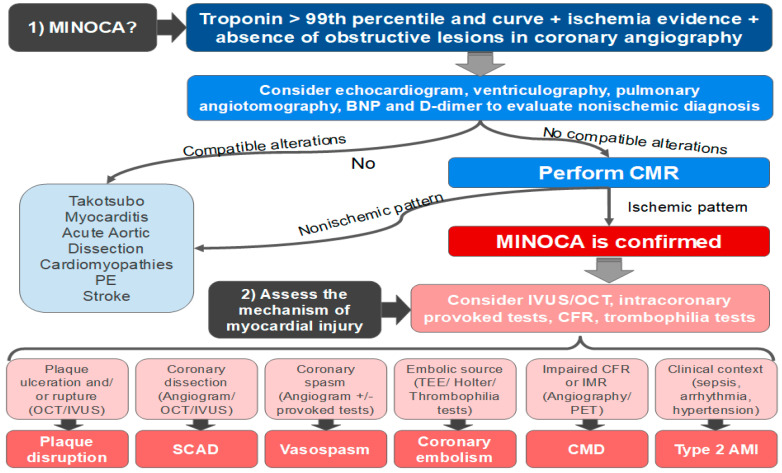
Flowchart for the diagnosis of MINOCA. BNP = brain natriuretic peptide; CMD = coronary microvascular dysfunction; CFR = coronary flow reserve; IMR = microvascular resistance indices; IVUS = intravascular ultrasound; MINOCA = myocardial infarction with non-obstructive coronary arteries; AMI = acute myocardial infarction; CMR = cardiac magnetic resonance; OCT = optical coherence tomography; PE = pulmonary embolism; PET = positron emission tomography; SCAD = spontaneous coronary artery dissection.

**Figure 11 jcm-11-05497-f011:**
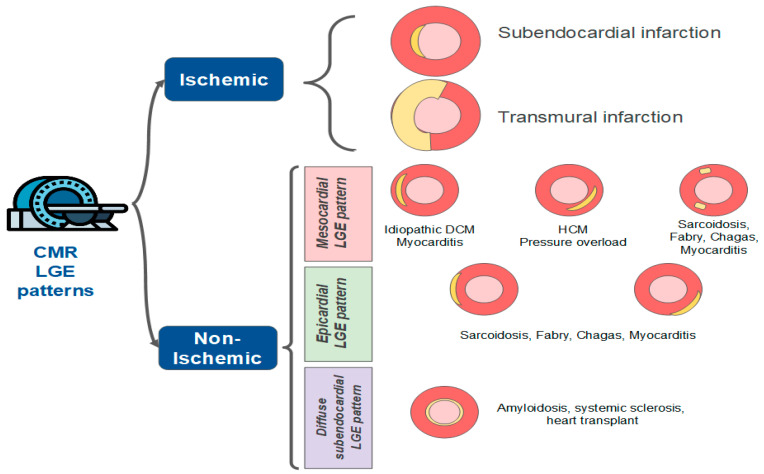
Late-gadolinium enhancement (LGE) patterns of myocardial injury. CMR: cardiac magnetic resonance; DCM: Dilated cardiomyopathy; HCM: Hypertrophic cardiomyopathy.

**Figure 12 jcm-11-05497-f012:**
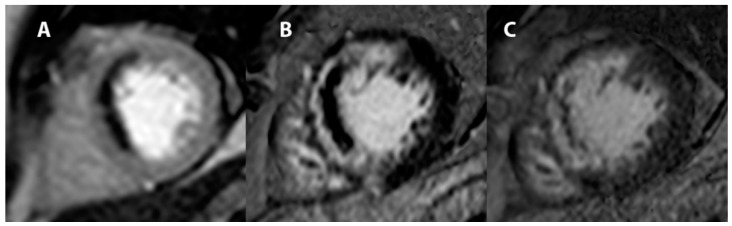
Cardiac resonance magnetic (CRM) showing first-pass perfusion (30 s) (**A**), early enhancement (4 min) (**B**), and late enhancement (15 min) (**C**), demonstrating impaired coronary microcirculation.

**Figure 13 jcm-11-05497-f013:**
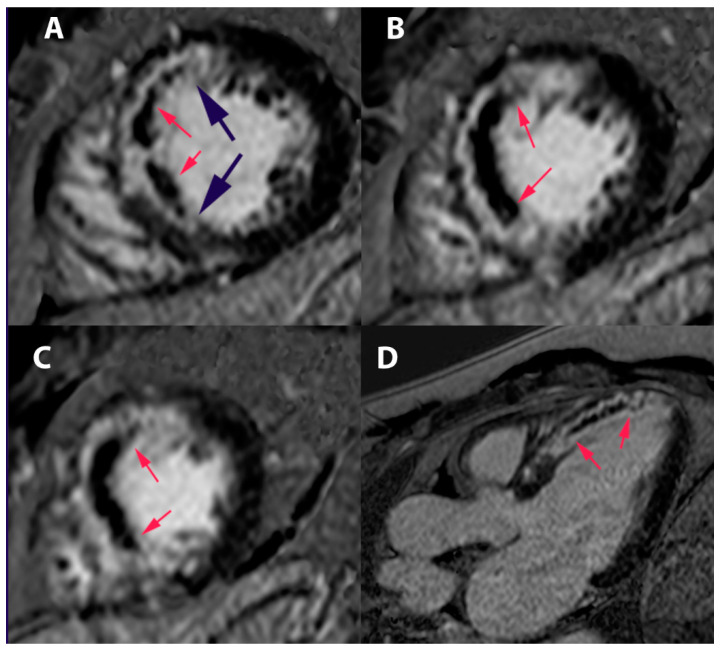
Cardiac resonance magnetic (CRM) showing delayed enhancement on left ventricular images: (**A**) short axis (base); (**B**) short axis (medium); (**C**) short axis (apical); and (**D**) long axis of the left ventricular outflow tract. Black arrows indicate anteroseptal myocardial infarction area of the left ventricle (white area). Red arrows indicate no reflow (microvascular obstruction) image.

**Figure 14 jcm-11-05497-f014:**
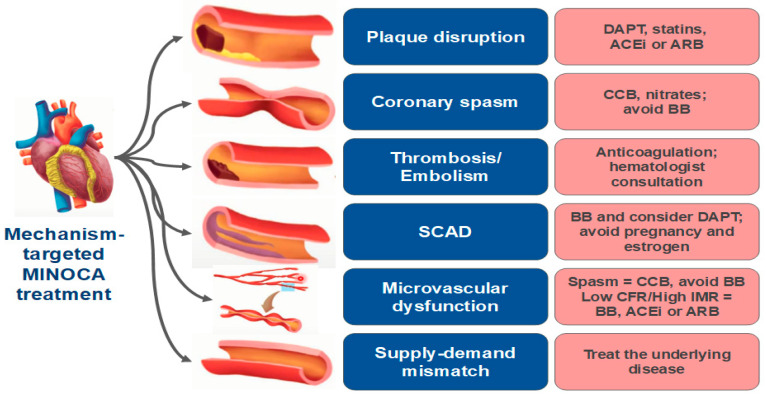
Mechanism-targeted MINOCA treatment. ACEi = angiotensin-converting enzyme inhibitor; ARB = angiotensin receptor blockers; BB beta-blocker; CCB = calcium channel blocker; CFR = coronary flow reserve; DAPT = dual antiplatelet therapy; MINOCA = myocardial infarction with non-obstructive coronary arteries; IMR = microvascular resistance indices; SCAD = spontaneous coronary artery dissection.

**Table 1 jcm-11-05497-t001:** Diagnostic criteria for myocardial infarction with non-obstructive coronary arteries.

AMI (a) Rise or fall of troponin with at least 1 value > the 99th percentile upper reference limit (b) At least 1 evidence of acute ischemia: (i) Symptoms (e.g., chest pain) (ii) New ischemic ECG changes or pathological Q waves (iii) New imaging finding compatible with ischemia (e.g., regional wall motion abnormality) (iv) Coronary thrombus on angiography or autopsy
2. Non-obstructive coronary lesions on angiography (a) Absence of coronary stenosis > 50% in any major epicardial vessel (b) The presence of stenosis < 50% do not configure obstructive coronary arteries
3. No specific alternate diagnosis (a) Diagnosis of non-ischemic subjacent causes such as myocarditis, pulmonary embolism, TCM and sepsis excludes the final diagnosis of MINOCA.

AMI = acute myocardial infarction; ECG = electrocardiogram; TCM = Takotsubo cardiomyopathy; MINOCA = myocardial infarction with non-obstructive coronary arteries.

**Table 2 jcm-11-05497-t002:** Coronary embolism diagnostic criteria.

Major criteria
Angiographic evidence of coronary artery embolism and thrombosis without atherosclerotic components Concomitant coronary artery embolization at multiple sites * Concomitant systemic embolization without left ventricular thrombus attributable to acute myocardial infarction
Minor criteria
<25% stenosis on coronary angiography, except for the culprit lesion Evidence of an embolic source based on transthoracic echocardiography, transesophageal echocardiography, computed tomography, or CMR Presence of embolic risk factors: atrial fibrillation, cardiomyopathy, rheumatic valve disease, prosthetic heart valve, patent foramen ovale, atrial septal defect, history of cardiac surgery, infective endocarditis, or hypercoagulable state
Definite CE
Two or more major criteria, or One major criterion plus ≥2 minor criteria, or Three minor criteria
Probable CE
One major criterion plus one minor criterion, or Two minor criteria
A diagnosis of CE should not be made if there is:
Pathological evidence of atherosclerotic thrombus History of coronary revascularization Coronary artery ectasia Plaque disruption or erosion detected by intravascular ultrasound or optic coherence tomography in the proximal part of the culprit lesion

CMR = cardiac magnetic resonance; CE = coronary artery embolism. * Indicates multiple vessels within one coronary artery territory or multiple vessels in the coronary tree.

**Table 3 jcm-11-05497-t003:** Diagnostic criteria for the microvascular angina (MVA). Definitive MVA if all 4 criteria are present. Suspected MVA if symptoms of ischemia with no obstructive coronary artery disease are present (criteria 1 and 2) but only objective evidence of myocardial ischemia (criteria 3) or evidence of impaired coronary microvascular function (criteria 4) alone.

Symptoms of myocardial ischemia (a) Effort and/or rest angina (b) Angina equivalents (i.e., shortness of breath)
2. Absence of obstructive CAD (<50% diameter reduction or FFR by >0.80) by (a) CCTA (b) Invasive coronary angiography
3. Objective evidence of myocardial ischemia (a) Ischemic ECG changes during an episode of chest pain (b) Stress-induced chest pain and/or ischemic ECG changes in the presence or absence of transient/reversible abnormal myocardial perfusion and/or wall motion abnormality
4. Evidence of impaired coronary microvascular function (a) Impaired coronary flow reserve (cut-off values depending on methodology use between ≤2.0 and ≤2.5) (b) Abnormal coronary microvascular resistance indices (i.e., IMR > 25) (c) Coronary microvascular spasm, defined as reproduction of symptoms, ischemic ECG shifts but no epicardial spasm during acetylcholine testing. (d) Coronary slow flow phenomenon, defined as TIMI frame count >25.

CAD = coronary artery disease; ECG = electrocardiogram; CCTA = coronary computed tomography angiography; FFR = fractional flow reserve; IMR = index of microcirculatory resistance; TIMI = thrombolysis in myocardial infarction.

**Table 4 jcm-11-05497-t004:** Differential diagnosis of Takotsubo and tests results.

Tests	Acute Myocardial Infarction	Takotsubo Cardiomyopathy	Myocarditis
Transthoracic Echocardiogram	Segmental ventricular dysfunction (respect coronary anatomy)	Regional ventricular dysfunction (does not respect coronary anatomy)	Segmental, regional or global ventricular dysfunction
Coronary angiography	Culprit obstructive lesion or absence of obstructive lesion (MINOCA)	Absence of obstructive lesion (more common) or not culprit obstructive lesion	Absence of obstructive lesion (more common) or not culprit obstructive lesion
Cardiac Magnetic Resonance	Subendocardial or transmural late gadolinium enhancement (LGE)	LGE absent	Patch and epicardial LGE are common than subendocardial or transmural

**Table 5 jcm-11-05497-t005:** Diagnostic cardiac magnetic resonance (CMR) criteria for myocarditis.

In the setting of clinically suspected myocarditis, CMR findings are consistent with myocardial inflammation, if at least two of the following criteria are present: Regional or global myocardial signal intensity increase in T2-weighted edema images Increased global myocardial early gadolinium enhancement ratio between myocardium and skeletal muscle in gadolinium-enhanced T1-weighted images There is at least one focal lesion with non-ischemic regional distribution in inversion recovery-prepared gadolinium-enhanced T1-weighted images LGE
A CMR study is consistent with myocyte injury and/or scar caused by myocardial inflammation if criterion 3 is present
A repeat CMR study between 1 and 2 weeks after the initial CMR study is recommended if: None of the criteria are present, but the onset of symptoms has been very recent and there is strong clinical evidence for myocardial inflammation One of the criteria is present
The presence of LV dysfunction or pericardial effusion provides additional, supportive evidence for myocarditis

CMR = cardiac magnetic resonance; LGE = large gadolinium enhancement; LV = left ventricle.

## Data Availability

Not applicable.

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
