# Peer review of "MINOCA: One Size Fits All? Probably Not—A Review of Etiology, Investigation, and Treatment"

_jcm, 2022, doi:10.3390/jcm11195497_

Round 1

Reviewer 1 Report

Manuscript ID: jcm-1903665 Title: MINOCA: one size fits all? Probably not. A review of etiology, investigation and treatment

Comments to the authors

Manuscript is well written and reviewed.

In the 2019 ESC guidelines, SPECT is not on the list of recommended CMD measurement modalities. On the other hand, the guidelines also include revisions to CMD assessments, and the recommended level of guidewire-based CFR and IMR measurements has been raised from Class II.B to Class II.A. You should make a description about the evaluation and efficacy of CMD by guidewire-based CFR and IMR measurement.

Author Response

Reviewer 1

Manuscript is well written and reviewed.

Point 1: In the 2019 ESC guidelines, SPECT is not on the list of recommended CMD measurement modalities. On the other hand, the guidelines also include revisions to CMD assessments, and the recommended level of guidewire-based CFR and IMR measurements has been raised from Class II.B to Class II.A. You should make a description about the evaluation and efficacy of CMD by guidewire-based CFR and IMR measurement.

Response 1: We appreciate your comments. Excellent observation. We have added information regarding CMD assessment by guidewire-based CFR and IMR measurement.

“The index of microcirculatory resistance (IMR) was firstly developed by Fearon et al. and is calculated from estimates of maximal distal coronary flow during hyperemia and pressure [85]. Ng et al showed that IMR is superior to CRF for assessing the coronary microcirculation by virtue of being more reproducible and less hemodynamically dependent than CFR [86] since it is not dependent on resting values. Moreover, IMR is not affected by epicardial stenosis severity [87]. Other indices can be used to assess CMD, such as hyperemic microvascular resistance [88], resistive reserve ratio [89], and microvascular resistance reserve (MRR) [90].

Recently, the novel technique to quantify absolute coronary flow and resistance through intracoronary continuous thermodilution has been developed. Morris et al. has demonstrated that this new method provides a comprehensive coronary physiological assessment of flow, pressure and resistance, across the entire coronary circulation, without the need for additional hardware, catheters, wires, or infusions [91].

However, the body of evidence concerning coronary flow and flow reserve measurement among the MINOCA population is currently limited. Similarly, the role and the clinical implications of continuous thermodilution-derived indexes within MINOCA patients are not yet established [92].”

Reviewer 2 Report

I read with interest the manuscript entitled “MINOCA: one size fits all? Probably not. A review of etiology, investigation and treatment “.  In their review, the authors sought to describe the epidemiology, definition, etiologies and management of MINOCA, which is a large entity.

General comments:

-       Literature is already abundant on this topic and one could say that it lacks a little novelty. Nevertheless, the topic is of interest the manuscript reads well.

-       There are few grammatical mistakes that should be corrected. Indeed, linking words are used to often. The manuscript could benefit from language editing.

-       All figures are mentioned in the text but are absent from the PDF file. Twelve figures might be a little too much. The authors could think about merging some of them in order not to overload the manuscript with too many figures.

Specific comments:

-       “The diagnosis of SCAD is usually possible with coronary angiography alone, but in- 161
travascular imaging such as IVUS or OCT is paramount for detecting more challenging 162
SCAD cases (Figure 3). However, while OCT images in SCAD are characteristic, IVUS 163
images require closer scrutiny to discriminate between plaque disruption and SCAD, 164
given the lower spatial resolution of IVUS [45].” OCT has better spatial resolution but is at increased risk of SCAD extension. As it requires quick and large contrast injection, SCAD extension can occur when using OCT to document SCAD. In addition, the more recent IVUS HD has better spatial resolution, which helps with the diagnostic of SCAD.

-       “Limitations of CCTA include low spatial resolution for small 170
vessels and with a diameter < 2.5 mm and for the middle and distal portions of the coro- 171
nary arteries, motion artifact, and unknown sensitivity and specificity [45].” Therefore CCTA can have its place for proximal SCAD but very few for mid and distal SCAD which is the preferred location for SCAD.

-       Of interest, myocardial bridging, per se, is unlikely to cause MINOCA. However, it 180
can predispose the affected artery to spasm. » Please add a reference if this has been described

-       “CAS results from the interaction of two components: (1) a usually localized but some- 182
times diffuse abnormality of a coronary artery that makes it hyperreactive to vasoconstric- 183
tor stimuli, and (2) a vasoconstrictor stimulus capable of inducing spasm at the level of 184
the hyperreactive coronary segment.” This is not exactly true. There is no systematic proven localized/diffuse coronary abnormality that makes it hyperreactive.

-       « Angina occurs predominantly at rest and may occur from midnight to early morn- 213
ing; » May occur MORE FREQUENTLY from midnight to earluy morning.

-       Although accurate, provocative tests are not promptly available, and concerns on 226
risks may exist.” I don’t really understand what the authors mean with this sentence?

-       « The risk of death is low, but the incidence of cardiac arrhythmias is rela- 227
tively high (6.8%) [60]. A contemporary study analyzed 921 patients undergoing intracor- 228
onary acetylcholine testing, and no deaths or serious complications were reported [61]. 229
Also, only 1% of patients had any kind of complications, namely non-sustained ventricu- 230
lar tachycardia, fast paroxysmal atrial fibrillation, bradycardia with hypotension and cath- 231
eter-induced vasospasm.” Therefore I would synthetize saying that complications of provocative test may exist but that none are associated with increased morbi mortality.

-       “Microvascular spasm” was not described by the authors and warrants discussion

-       The dysfunction affects only these vessels and it is character- 284
ized by reduced coronary flow reserve (CFR)”. The authors should explain the difference between FFR CFR and IMR and therefore mention that CFR is reduced but IMR elevated with normal FFR. Indeed, CFR can be reduced due to low FFR with normal microcirculation.

-       “Microvascular dysfunction plays a major role in determining myocardial ischemia in 305
many cardiovascular conditions, not only in MINOCA, but also in non-ischemic cardio- 306
myopathies, Takotsubo syndrome and heart failure » This was never really invasively proven

-       « Diagnosis of microvascular dysfunction includes invasive methods such as CFR and 329
index of microvascular resistance, and non-invasive methods such as positron emission 330
tomography (PET), CMR, and Doppler echocardiograph” Absolute blood flow is another method of microcirculation invasive evaluation

-       « Both IVUS and OCT are safe procedures. A single-center study evaluated adverse 478
events in patients undergoing invasive imaging during coronary catheterization [106]. In- 479
vasive imaging-related complications were rare, did not differ between the two imaging 480
methods (0.5-0.6%). No major adverse events, prolongation of hospital stay, or permanent 481
patient harm was observed. » During SCAD, a wire has to be introduced in the lumen and can end up in the false lumen inducing SCAD extension + OCT induced SCAD extension. The authors should emphasized that this should be used when diagnosis is uncertain.

-       PCI in SCAD patients has been associated to more adverse events, with propagation 544
of the hematoma, and worse final results. A retrospective study found PCI failure rate of 545
53% and did not protect against target vessel revascularization or recurrent SCAD [42]. 546
Therefore, coronary revascularization is recommended only for patients at high risk due 547
to involvement of multivessel severe proximal dissections or of the left main artery or the 548
ostial left anterior descending artery, hemodynamic instability, or refractory arrhythmia 549
[24].” SCAD opening with cutting/scoring balloon is now better admitted as compared with PCI and stent implantation. This can be discussed.

-       MINOCA is not benign and has comparable outcomes with AMI-CAD” really comparable?

Author Response

Reviewer 2

 General comments:

Point 1: Literature is already abundant on this topic and one could say that it lacks a little novelty. Nevertheless, the topic is of interest the manuscript reads well.

Response 1: Thanks for the comments. In fact, this topic has been much explored in the literature recently. On the other hand, new studies on MINOCA have been recently published.

Point 2: There are few grammatical mistakes that should be corrected. Indeed, linking words are used to often. The manuscript could benefit from language editing.

Response 2: Thanks for the observation. We will send the article for English review.

Point 3: All figures are mentioned in the text but are absent from the PDF file. Twelve figures might be a little too much. The authors could think about merging some of them in order not to overload the manuscript with too many figures.

Response 3: MINOCA is a very broad topic. We decided to create several figures to make the reading more didactic to the reader.

Specific comments:

Point 4: “The diagnosis of SCAD is usually possible with coronary angiography alone, but intransvascular imaging such as IVUS or OCT is paramount for detecting more challenging SCAD cases (Figure 3). However, while OCT images in SCAD are characteristic, IVUS images require closer scrutiny to discriminate between plaque disruption and SCAD, given the lower spatial resolution of IVUS [45].”

OCT has better spatial resolution but is at increased risk of SCAD extension. As it requires quick and large contrast injection, SCAD extension can occur when using OCT to document SCAD. In addition, the more recent IVUS HD has better spatial resolution, which helps with the diagnostic of SCAD.

Response 4: Thank you for pointing this out. We agree with this and have incorporated your suggestion throughout the manuscript. We have added this information to the manuscript.

“However, OCT could further aggravate the dissection or exacerbate a new intimate tear due to contrast injection. In addition, the more recent high-definition IVUS (HD IVUS) has better spatial resolution, which helps with the diagnosis of SCAD [46].”

Point 5: “Limitations of CCTA include low spatial resolution for small vessels and with a diameter < 2.5 mm and for the middle and distal portions of the coronary arteries, motion artifact, and unknown sensitivity and specificity [45].”

Therefore CCTA can have its place for proximal SCAD but very few for mid and distal SCAD which is the preferred location for SCAD.

Response 5: Thank you very much for your observation. We took the liberty of adding this sentence to the manuscript.

Point 6: “Of interest, myocardial bridging, per se, is unlikely to cause MINOCA. However, it can predispose the affected artery to spasm.”

Please add a reference if this has been described.

Response 6: Thank you very much. We have added references to manuscript sentences.

“Of interest, myocardial bridging, per se, is unlikely to cause MINOCA [54]. However, it can predispose the affected artery to spasm [55,56].

Proposed mechanisms to constitute the substrate for CAS susceptibility include: (1) endothelial dysfunction, and (2) primary hyperreactivity of vascular smooth muscle cells [57].”

Point 7: “CAS results from the interaction of two components: (1) a usually localized but sometimes diffuse abnormality of a coronary artery that makes it hyperreactive to vasoconstrictor stimuli, and (2) a vasoconstrictor stimulus capable of inducing spasm at the level of the hyperreactive coronary segment.”

This is not exactly true. There is no systematic proven localized/diffuse coronary abnormality that makes it hyperreactive.

Response 7: Thanks for this observation. We have deleted this passage from the manuscript and added a reference to the subsequent sentence.

“Proposed mechanisms to constitute the substrate for CAS susceptibility include: (1) endothelial dysfunction, and (2) primary hyperreactivity of vascular smooth muscle cells [57].”

Point 8: “Angina occurs predominantly at rest and may occur from midnight to early morning“

May occur MORE FREQUENTLY from midnight to early morning.

Response 8: Thanks. We made the correction in the manuscript.

“Angina occurs predominantly at rest and may occur more frequently from midnight to early morning.”

Point 9: “Although accurate, provocative tests are not promptly available, and concerns on risks may exist.”

I don’t really understand what the authors mean with this sentence?

Response 9: Thanks for the observation. We have corrected the sentence as follows:

“Although accurate [66], provocative tests for CAS are associated with risks.”

Point 10: “The risk of death is low, but the incidence of cardiac arrhythmias is relatively high (6.8%) [60]. A contemporary study analyzed 921 patients undergoing intracoronary acetylcholine testing, and no deaths or serious complications were reported [61]. Also, only 1% of patients had any kind of complications, namely non-sustained ventricular tachycardia, fast paroxysmal atrial fibrillation, bradycardia with hypotension and catheter-induced vasospasm.”

Therefore I would synthetize saying that complications of provocative test may exist but that none are associated with increased morbimortality.

Response 10: Thank you for pointing this out. We have added this information to the sentence.

“Although complications related to provocative tests may exist, none of them are associated with increased morbidity and mortality.”

Point 11: “Microvascular spasm” was not described by the authors and warrants discussion.

Response 11: Very good observation. We have added this topic to the manuscript.

“Therefore, CAS can occur at the level of the epicardial arteries as well as in the coronary microcirculation. Current standardized diagnostic criteria for microvascular spasm include reproduction of the patient's angina symptoms and ischemic ECG changes in the absence of epicardial spasm during intracoronary spasm provocation testing using, for example, acetylcholine [53]. Of note, it is important to mention that it is difficult to identify the mechanism of microvascular dysfunction that triggers microvascular angina. Therefore it is essential to distinguish between an impaired microcirculatory vasodilatory capacity, which can be diagnosed by measuring coronary flow reserve or microvascular resistance, and microvascular spasm determined by intracoronary acetylcholine administration.”

Point 12: “The dysfunction affects only these vessels and it is characterized by reduced coronary flow reserve (CFR)”.

The authors should explain the difference between FFR, CFR and IMR and therefore mention that CFR is reduced, but IMR elevated with normal FFR. Indeed, CFR can be reduced due to low FFR with normal microcirculation.

Response 12: Excellent suggestion. We have added the following to the manuscript:

“CFR is an invasive method that allows an integrated measurement of flow through the large epicardial arteries and coronary microcirculation, but once severe obstructive disease of the epicardial arteries is ruled out, reduced CFR is a marker of CMD. CFR is the ratio of hyperemic blood flow divided by resting blood flow and can be calculated using thermodilution or Doppler flow velocity. Overall, the prognostic value of CFR used a cutoff value <2.0 [73]. The index of microcirculatory resistance (IMR) is calculated as the product of distal coronary pressure at maximal hyperemia multiplied by the hyperemic mean transit time. IMR ≥25 is representative of microvascular dysfunction [74-76]. Flow-limiting obstructive coronary artery disease can be assessed using Fractional Flow Reserve (FFR), which is the ratio of mean distal coronary pressure to mean aortic pressure at maximal hyperemia (abnormal FFR is defined as ≤0.80) [73]. FFR values >0.8, CFR ≥2.0 and IMR <25 represent absence of CMD and after vasoactive stimuli with acetylcholine with absence or reduction of coronary diameter <90%, without angina and lack of ischemic ECG changes it is interpreted as pain non-cardiac and the opposite changes in the test allow the diagnosis of VSA. The FFR values >0.8, CFR <2.0 and IMR ≥25 represent the presence of CMD and after a vasoactive stimuli with acetylcholine with absence or reduction of coronary diameter <90%, without angina and lack of ischemic ECG changes it is interpreted as microvascular angina and the opposite test result allows diagnosis of microvascular angina and VSA.”

Point 13: “Microvascular dysfunction plays a major role in determining myocardial ischemia in many cardiovascular conditions, not only in MINOCA, but also in non-ischemic cardiomyopathies, Takotsubo syndrome and heart failure”

This was never really invasively proven

Response 13: Thanks for the observation. In fact, we have looked at the literature and have not really found any scientific evidence to support this claim, although it is often described in several manuscripts. Therefore, we decided to remove this sentence from the text and the corresponding figure.

Point 14: “Diagnosis of microvascular dysfunction includes invasive methods such as CFR and index of microvascular resistance, and non-invasive methods such as positron emission tomography (PET), CMR, and Doppler echocardiograph”

Absolute blood flow is another method of microcirculation invasive evaluation

Response 14: Thanks for the observation. We have added this information to the manuscript.

“Assessment of microvascular dysfunction includes invasive methods such as CFR, index of microvascular resistance (IMR), and absolute coronary blood flow measured, and non-invasive methods such as positron emission tomography (PET), CMR, and Doppler echocardiography.

Recently, the novel technique to quantify absolute coronary flow and resistance through intracoronary continuous thermodilution has been developed. Morris et al. has demonstrated that this new method provides a comprehensive coronary physiological assessment of flow, pressure and resistance, across the entire coronary circulation, without the need for additional hardware, catheters, wires, or infusions [91].”

Point 15: “Both IVUS and OCT are safe procedures. A single-center study evaluated adverse events in patients undergoing invasive imaging during coronary catheterization [106]. Invasive imaging-related complications were rare, did not differ between the two imaging methods (0.5-0.6%). No major adverse events, prolongation of hospital stay, or permanent patient harm was observed.”

During SCAD, a wire has to be introduced in the lumen and can end up in the false lumen inducing SCAD extension + OCT induced SCAD extension. The authors should emphasized that this should be used when diagnosis is uncertain.

Response 15: Thanks again for this observation. We have added this information to the manuscript.

“However, OCT has only been recommended when the diagnosis of SCAD is uncertain, since the introduction of the guidewire into the lumen can end up in the false lumen inducing SCAD extension.”

Point 16: “PCI in SCAD patients has been associated to more adverse events, with propagation of the hematoma, and worse final results. A retrospective study found PCI failure rate of 53% and did not protect against target vessel revascularization or recurrent SCAD [42]. Therefore, coronary revascularization is recommended only for patients at high risk due to involvement of multivessel severe proximal dissections or of the left main artery or the ostial left anterior descending artery, hemodynamic instability, or refractory arrhythmia [24].”

SCAD opening with cutting/scoring balloon is now better admitted as compared with PCI and stent implantation. This can be discussed.

Response 16: Thank you for pointing this out. We have added this information to the sentence.

“Although the experience with cutting balloon (CB) angioplasty for SCAD is limited, recent studies have compared CB with PCI for SCAD [134,135]. CB angioplasty can fenestrate the false lumen to allow communication and back-bleed of intramural hematoma into the true lumen.”

Point 17: “MINOCA is not benign and has comparable outcomes with AMI-CAD”

Really comparable?

Response 17: Thanks for the question. We conclude that MINOCA is not benign and has comparable outcomes with AMI-CAD based on the results of the VIRGO (Results From the Variation in Recovery: Role of Gender on Outcomes of Young AMI Patients) study. Safdar and colleagues have shown that although the characteristics of patients with MINOCA and their counterparts with AMI and CAD (AMI-CAD) were different, the mortality rates at 1 month (1.1% versus 0.6%, P=0.43) and 1 year (1.7 % versus 2.3%, P=0.68) were not statistically different [. Safdar B, Spatz ES, Dreyer RP, Beltrame JF, Lichtman JH, Spertus JA, Reynolds HR, Geda M, Bueno H, Dziura JD, Krumholz HM, D'Onofrio G. Presentation, clinical profile, and prognosis of young patients with myocardial infarction with nonobstructive coronary arteries (MINOCA): results from the VIRGO Study. J Am Heart Assoc 2018;7:e009174. DOI: 10.1161/JAHA.118.009174].

We have added this information to the manuscript.

“MINOCA is not benign and has comparable outcomes with AMI-CAD, according to the findings of the VIRGO study [2].”
